

# PopAlu: population-scale detection of Alu polymorphisms

Yu Qian[1], Birte Kehr[2] and Bjarni V. Halldórsson[2,3]

[1] Bioinformatics Research Center, Aarhus University, Aarhus, Denmark
[2] deCODE genetics/Amgen, Reykjavík, Iceland
[3] Institute of Biomedical and Neural Engineering, School of Science and Engineering, Reykjavik University, Reykjavík, Iceland

## ABSTRACT

Alu elements are sequences of approximately 300 basepairs that together comprise more than 10% of the human genome. Due to their recent origin in primate evolution some Alu elements are polymorphic in humans, present in some individuals while absent in others. We present PopAlu, a tool to detect polymorphic Alu elements on a population scale from paired-end sequencing data. PopAlu uses read pair distance and orientation as well as split reads to identify the location and precise breakpoints of polymorphic Alus. Genotype calling enables us to differentiate between homozygous and heterozygous carriers, making the output of PopAlu suitable for use in downstream analyses such as genome-wide association studies (GWAS). We show on a simulated dataset that PopAlu calls Alu elements inserted and deleted with respect to a reference genome with high accuracy and high precision. Our analysis of real data of a human trio from the 1000 Genomes Project confirms that PopAlu is able to produce highly accurate genotype calls. To our knowledge, PopAlu is the first tool that identifies polymorphic Alu elements from multiple individuals simultaneously, pinpoints the precise breakpoints and calls genotypes with high accuracy.

## INTRODUCTION

Population-wide identification of variation has recently become possible through falling costs of DNA sequencing. The list of whole-genome sequencing projects with large numbers of individuals is constantly growing (*Gudbjartsson et al., 2015*; *Genome of the Netherlands Consortium, 2014*; *The 1000 Genomes Project Consortium, 2012*) due to its potential to characterize genetic variation and advance medical research.

Alu elements are a substantial source of structural variation in human genomes. This class of active mobile elements (MEs) is abundant in all primate species and comprises more than 10% of the human genome (*Cordaux & Batzer, 2009*). The Alu sequences are approximately 300 bp long with a dimeric structure separated by a short A-rich region, each monomer being derived from the 7SL RNA gene.

Although Alu elements do not encode genes, many studies suggest their functional importance. Alu elements are recognized to affect protein synthesis at the transcriptional

Corresponding author
Bjarni V. Halldórsson,
bjarni.halldorsson@decode.is

and post-transcriptional level (*Sorek, Ast & Graur, 2002*; *Kelley et al., 2014*) as well as DNA methylation (*De Andrade et al., 2011*) and other cellular processes (*Deininger & Batzer, 1999*). Furthermore, they are thought to be major drivers of genome evolution (*Hormozdiari et al., 2013*; *Salem et al., 2003*) and assist in the creation of structural variation (*Wang et al., 2006*). The importance of Alu elements is further highlighted by the potential association with genetic instability, one of the principal causative factors in many disorders including cancer (*Deininger & Batzer, 1999*; *Zhang et al., 2011*; *Helman et al., 2014*).

Alu elements have been inserted into the human genome at more than one million locations over the last 65 million years (mya). The majority of amplifications happened early in primate evolution. The estimated current rate of Alu retrotransposition is approximately one per generation, which is at least 100-fold slower than at the peak of amplification that occurred 30–50 mya ago (*Batzer & Deininger, 2002*; *Kapitonov & Jurkal, 1996*; *Witherspoon et al., 2013*).

The evolutionary history of Alu elements led to two broad categories, to *fixed* Alu elements and *polymorphic* Alu elements. Fixed Alu elements are present in the entire population and, thus, are presumably evolutionarily older. Their locations are largely known from the reference genome. In contrast, polymorphic Alu elements appear only in a subset of the population and, hence, are likely the result of more recent retrotransposition events. The reference genome contains only some of these newly inserted Alus and the locations of many other polymorphic Alu elements are still unknown.

Identification of polymorphic Alu elements from sequencing data can be divided into two problems, the detection of *Alu deletions* and the detection of *Alu insertions*. The goal of the Alu deletion problem is to find Alu elements present in the reference but not in a sequenced genome. The goal of the Alu insertion problem is to find Alu elements missing in the reference but present in a sequenced genome. We acknowledge that even the Alu elements found by solving the deletion problem have most likely been inserted during evolution. In addition to the discovery of polymorphic Alu loci, we consider genotyping as an integral part of both problems as it is typically necessary for downstream analyses. Unlike the discovery, which distinguishes only between the two states, 'fixed' or 'polymorphic', genotyping classifies individuals into three genotypes, 'non-carrier', 'heterozygous carrier' or 'homozygous carrier'.

While a large number of methods have been developed to determine other types of variation from sequencing data, such as SNPs and small indels, comparatively fewer methods have been developed for finding structural variation and in particular Alu polymorphisms. Notable exceptions are Alu-Detect (*David, Mustafa & Brudno, 2013*), VariationHunter (*Hormozdiari et al., 2010*; *Hormozdiari et al., 2013*) RetroSeq (*Keane, Wong & Adams, 2013*), Tangram (*Wu et al., 2014*), and Mobster (*Thung et al., 2014*). These methods focus mainly on the detection of Alu insertions, and generally follow a three-step analysis. First, they identify fragments (reads or read pairs) that indicate the occurrence of an Alu insertion. Next, they cluster these fragments along the genome, such that each cluster includes a potential insertion. Last, for each sequenced genome and at each cluster, they calculate a likelihood that an Alu element has actually been inserted given

the set of fragments. These steps are similar to approaches implemented in another class of programs that discover novel sequence insertions (*Kehr, Melsted & Halldórsson, 2015*; *Rizk et al., 2014*; *Hajirasouliha et al., 2010*). But unlike novel sequences, the sequence of Alu polymorphisms is known and repetitive, i.e., has inserted at more than a single location in the genome, which is why the novel sequence discovery programs do not detect mobile element polymorphisms.

In this paper, we describe the tool *PopAlu* for *pop*ulation-wide detection of *Alu* polymorphisms. PopAlu is the successor of our previous tool PAIR (*Sveinbjörnsson & Halldórsson, 2012*) with a number of improvements and an extension to handle many individuals simultaneously. It follows the three-step approach and starts by identifying read pairs indicating an Alu polymorphism. As opposed to most other methods, PopAlu constructs clusters in the second step using read pairs from many individuals simultaneously. Pooling of data across many individuals increases detection power even for polymorphisms of low frequency. Further, PopAlu pinpoints the precise insertion breakpoints instead of reporting only approximate locations. In the last step, PopAlu applies a probabilistic approach for calling genotypes. It can differentiate between homozygous and heterozygous calls, while many other tools either do not report heterozygous calls or make calls simply based on the counts of supporting fragments. Our implementation of PopAlu is easy to use in that it is almost parameter-free—most of the parameters are automatically inferred from the input data—and it is a stand-alone package implemented using the SeqAn C++ library (*Döring et al., 2008*) without any further requirements of external tools.

## METHODS

The input of PopAlu is a reference genome and a binary alignment (BAM) file of paired-end sequencing reads of a donor individual (or a set of individuals).

### Definitions

A read pair $r$ has a left read $r^L$ and right read $r^R$, which are mates to each other, denoted as mate($r^L$) = $r^R$ and mate($r^R$) = $r^L$. We use $r^N$ to denote either a left or a right read when the relative position in the pair is not relevant. If $r^N$ is mapped to the reference genome, we use begin($r^N$) and end($r^N$) to represent its start and end position in the mapped reference genome. We say that a read is *concordant* if the two ends are mapped to opposite strands on the same chromosome within a close distance of each other and otherwise *discordant*. We define the insert length of a read pair $r$, measured with respect to the reference genome, as $Y(r) = \text{end}(r^R) - \text{begin}(r^L)$. We approximate the empirical distribution of $Y$ from all concordant read pairs, stratifying $Y$ by sequencing library. We let $E[Y]$ denote the mean of $Y$ and $\sigma(Y)$ the standard deviation. We refer to the bounds of the Alu region as *breakpoints*. Figure 1 shows an example Alu region bounded by a left breakpoint $AL$ and a right breakpoint $AR$.

### Alu deletion

Given the reference aligned sequencing data of a single individual and a set of known Alu elements in the reference genome, the objective of the Alu deletion problem is to
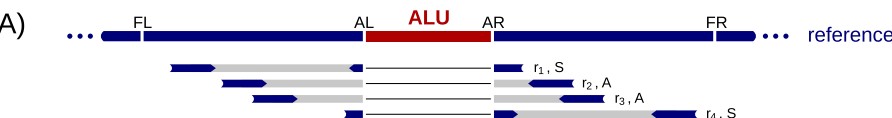

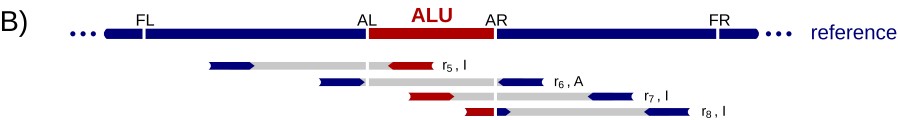

**Figure 1 Example read alignments at an Alu deletion site.** Arrows show read directions. The blue part of the reads can be mapped to the reference outside of the Alu and the red part can be mapped to the Alu. (A) shows example reads from a haplotype that carries allele $H_1$. (B) shows example reads from a haplotype that carries allele $H_0$. A heterozygote diploid can have reads shown in both (A) and (B).

examine each of the given Alu elements for the existence of an Alu deletion. The input set of Alu elements can be determined using various tools, e.g., RepeatMasker (http://www.repeatmasker.org). The main difficulty in this problem is to determine the genotypes of all individuals.

For each Alu element in the reference genome, we distinguish the two alleles $H_0$ and $H_1$; $H_0$ denotes the presence and $H_1$ denotes the absence of the Alu element with respect to the reference genome. Given these alleles, three autosomal genotypes are possible for an individual: homozygote Alu ($G_0$), heterozygote ($G_1$) and homozygote non-Alu ($G_2$). Our goal is to compute for each individual the relative likelihoods of the genotypes given the sequencing data.

Our algorithm considers each Alu element in the reference genome separately. Given an Alu sequence in the reference starting at position $AL$ and ending at $AR$, we restrict our attention to the region containing the Alu $[AL, AR]$ and flanking regions $[FL, AL]$ and $[AR, FR]$ on both sides of the Alu. We choose $|FR - AR| = |AL - FL| = E[Y] + 3\sigma(Y)$. From the aligned sequencing data, we then select the set of concordant read pairs $R$ such that for all $r \in R$ at least one read overlaps $[FL, FR]$. Given the set $R$ for an Alu element, our Alu deletion algorithm has two steps. In the first steps it classifies reads in $R$ and in the second step it computes relative likelihoods of the three genotypes.

### Read classification

There are primarily two signals indicating reads coming from a haplotype carrying the deletion allele $H_1$. The first is a read split into two parts, one part from each side of the Alu. The second is that the two reads in a read pair are aligned to different sides of the Alu with an increased insert length with respect to the reference genome. That is, the read pair's insert length follows the distribution $Y + l_{\text{Alu}}$ instead of the distribution $Y$, where $l_{\text{Alu}}$ is the length of the Alu sequence.

In order to distinguish these signals of the deletion allele $H_1$ from the ones that support allele $H_0$, we classify read pairs $r \in R$ into three types and remove those $r$ from $R$ that fulfill none of the types' criteria. In the end, we obtain a classification, $\mathcal{C}(R)$, that assigns each

read pair a type: $\text{type}(r) \in \{I, S, A\}$ for all $r \in R$. We use the notation $r \in X$ if a read pair $r$ is of type $X$.

- $I$ (Internal). A read pair $r$ is of type $I$ if either $r^L$ is mapped to $[FL, AL]$ and $r^R$ to $[AL, AR]$ or $r^L$ is mapped to $[AL, AR]$ and $r^R$ to $[AR, FR]$. Figure 1 displays three examples: $r_5$, $r_7$ and $r_8$.
- $S$ (Split). A read pair is of type $S$ if one of the reads in the pair is a split read. A read is a split read if a part of it is mapped to the left of $AL$ and the unmapped part aligns to the right of $AR$ or a part of it is mapped to the right of $AR$ and the unmapped part aligns to the left of $AL$. $r_1$ and $r_4$ in Fig. 1 are examples for $S$ reads.

   When identifying split reads we realign the unmapped part of a read using the Smith Waterman algorithm (*Smith & Waterman, 1981*). We allow mismatches and small gaps, but require a minimal alignment score and at least 20 bp aligned on each side of the Alu.
- $A$ (Across). A read pair $r$ is of type $A$ if $r^L$ is mapped to $[FL, AL]$ and $r^R$ is mapped to $[AR, FR]$. The insert length indicates its origin from $H_0$ or $H_1$. The read pairs $r_2$, $r_3$ and $r_6$ in Fig. 1 are examples for type $A$.

### Determining genotype

Based on the classification of read pairs, $\mathcal{C}(R)$, we compute a relative likelihood, $L$, of observing the reads given each genotype, $G_0$, $G_1$, and $G_2$. The following two paragraphs describe how we choose the likelihoods for observing each read pair given the alleles. Finally, we describe how we compute a joint likelihood for all read pairs given the three genotypes.

*Breakpoint overlapping reads.* Reads overlapping breakpoints, i.e., read pairs of types $I$ and $S$, give strong evidence for Alu polymorphisms. $S$ read pairs are most likely from a haplotype carrying the $H_1$ allele and $I$ read pairs are most likely from a haplotype carrying the $H_0$ allele. As we are only interested in the *relative* likelihoods of the data given the genotypes, we fix the likelihood, $L$, of such read pairs given the corresponding allele as 1:

$$L(r|H_0, r \in I) = L(r|H_1, r \in S) = 1$$

To account for misalignment or sequencing error, we set the likelihood of observing a read pair of type $I$ or $S$ given the other allele to a parameter $PE$, chosen as 0.001 in our experiments:

$$L(r|H_0, r \in S) = L(r|H_1, r \in I) = PE$$

*Spanning read pairs.* Read pairs spanning across an Alu, i.e., read pairs of type $A$, have either an insert length distribution $Y(r)$ if they come from a haplotype carrying the $H_0$ allele or they align $l_{Alu}$ further apart if they originate from a haplotype carrying the $H_1$ allele and, thus, have an insert length distribution $Y(r) + l_{Alu}$. Therefore, we can derive the likelihood of observing a read pair of type $A$ as:

$$L(r|H_0, r \in A) \sim Y(r)$$
$$L(r|H_1, r \in A) \sim Y(r) + l_{\text{Alu}}.$$

*Joint likelihood.* At a given Alu location, we assume that each read pair in the set $R$ is independent. The likelihood of the observed read pairs given the true genotype $G_g$, $g \in \{0,1,2\}$ and the read classification $\mathcal{C}(R)$ is, thus, as follows:

$$L(R|G_g, \mathcal{C}(R)) = \prod_{r \in R} L(r|G_g, r \in X)$$

$$= \prod_{r \in R} \{L(r|H_0, r \in X)P(H_0|G_g) + L(r|H_1, r \in X)P(H_1|G_g)\}$$

$$= \prod_{r \in R} \{L(r|H_0, r \in X)P(H_0|G_g) + L(r|H_1, r \in X)(1 - P(H_0|G_g))\} \qquad (1)$$

where $L(r|H_x)$ is given above and $P(H_x|G_g)$ is the probability of allele $H_x$ given genotype $G_g$. We have $P(H_0|G_0) = P(H_1|G_2) = 1$ for the homozygous genotype and use the estimate $P(H_0|G_1) = \frac{2 \cdot \|r\| + l_{Alu}}{4 \cdot \|r\| + l_{Alu}}$ for the heterozygous genotype assuming uniform sequencing coverage, and a read length of $\|r\|$, e.g., 100 bp. The intuition behind this estimate is that the probability for $H_0$ is relative to the length ratio of $H_0$ versus the sum of $H_0$ and $H_1$ where the length of $H_1$ is estimated as $2 \cdot \|r\|$ base pairs and $H_0$ as $2 \cdot \|r\| + l_{Alu}$ base pairs.

## Alu insertion

Detecting Alu insertions is a more difficult problem than detecting Alu deletions, as potential insertion positions (breakpoints) are not known *a priori*. When considering multiple individuals simultaneously, it is preferable to know the precise breakpoints shared by all carriers of the Alu insertion in order to make accurate genotype calls. Therefore, we first select reads that indicate the occurrence of an Alu insertion, cluster these reads by location per individual, and then combine the clusters of multiple individuals. Finally, we infer breakpoints for all candidate sites and insert a consensus Alu element *in silico* in order to apply the Alu deletion algorithm for genotype calling.

### Informative reads

There are mainly two signals indicating the presence of an Alu insertion. The first is a discordant read pair where one read is mapped to a known Alu region, cf. $r_5$ and $r_7$ in Fig. 2. We will refer to these discordant read pairs as $D$ read pairs. The second signal is a split read where only the part at one side of the breakpoint aligns to the reference genome, cf. $r_6$ and $r_8$ in Fig. 2. We denote these split reads as $C$ (clipped) reads, as they are often soft-clipped in the BAM files.

Our Alu insertion algorithm uses $D$ read pairs to identify candidate insertion sites, and $C$ reads to pinpoint the precise breakpoints. $D$ read pairs are widely used to infer approximate regions of Alu insertions (*David, Mustafa & Brudno, 2013*; *Keane, Wong & Adams, 2013*; *Hormozdiari et al., 2010*), whereas comparatively fewer tools utilize $C$ reads (*David, Mustafa & Brudno, 2013*).

### Insertion sites for a single individual

We start by scanning the BAM file of a single individual for $D$ read pairs, and classify the non-Alu reads of these pairs as *la* and *ra* reads. A *la* read maps in the forward orientation to the reference genome implying that its mapping location is to the left of an inserted Alu,

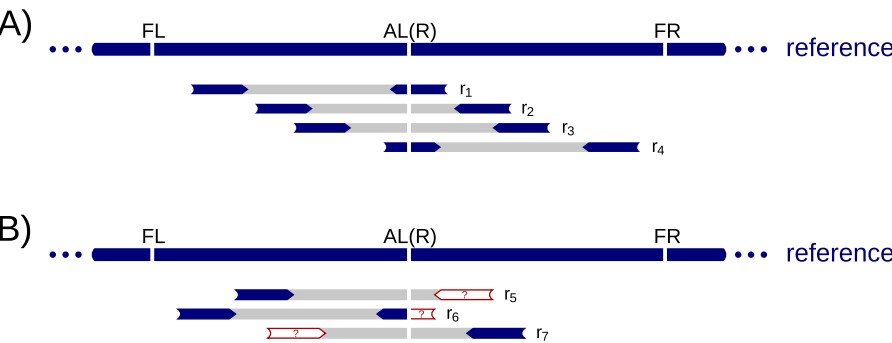

**Figure 2 Example read alignment at an Alu insertion site.** Arrows show read directions. The blue part of the reads can be mapped to the reference and the red parts are clipped or mapped somewhere else in the reference. (A) shows example reads from a non-Alu haplotype. (B) shows example reads from an Alu insertion haplotype. A heterozygote diploid can have reads shown in both (A) and (B).

cf. the blue end of $r_5$ in Fig. 2. A *ra* read maps in the reverse complemented orientation to the right of the Alu, cf. the blue end of $r_7$ in Fig. 2. Thus, both *la* and *ra* reads give partial information about the location of an Alu insertion.

Next, we iterate once over the reference genome to find all positions where $D$ read pairs cluster. We say that a position $p$ covers a *la* read $r^N$ if $p \in [\text{end}(r^N), \text{begin}(r^N) + E[Y] + 3\sigma(Y)]$ holds, and $p$ covers a *ra* read $r^N$ if $p \in [\text{end}(r^N) - E[Y] - 3\sigma(Y), \text{begin}(r^N)]$. We define $support(p)$ as the sum of *la* and *ra* reads covered by position $p$ and store all positions that fulfill $support(p) > n$, e.g., $n = 4$. This step can be done in $\mathcal{O}(m \log m + g)$ time, where $m$ is the total number of *la* and *ra* reads and $g$ is the size of the genome; First, we sort all *la* and *ra* reads by position, which takes time $\mathcal{O}(m \log m)$. Then, we move along the genome in linear time $\mathcal{O}(g)$ searching for positions passing $support(p) > n$ while updating two queues, of *la* and *ra* reads. In order to limit computation time, we only consider positions $p = \text{end}(ra)$ or $p = \text{end}(la)$ where $ra, la \in D$.

Finally, we create a set of non-overlapping regions $(b_i, e_i)$, $i = 0, 1, 2, \ldots$ from the set of positions $P$ that have a support greater than $n$. To ensure that at maximum one Alu insertion occurs in each $(b_i, e_i)$ we constrain the length of each region to $\ell$ (e.g., $\ell = 200$ bp) as follows. We traverse the positions $P$ in sorted order and initialize a first region $(b_0, e_0)$ with the first position $p_0 \in P$ as $b_0 = e_0 = p_0$. We iteratively extend a $(b_i, e_i)$ by updating $e_i$ for the next position $p_j \in P$ if $p_j - b_i < \ell$. Otherwise, we initialize a new region $(b_{i+1}, e_{i+1})$ with $b_{i+1} = e_{i+1} = p_j$ and continue with extending the new region.

At this point the breakpoint position is only approximate, we will use $C$ reads to refine the precise breakpoints in $(b_i, e_i)$ in later steps.

### Insertion sites for multiple individuals

To study multiple individuals simultaneously, we first find candidate insertion regions in each individual separately. Next, we pool the sets of candidate regions from all individuals, traverse the pooled set of regions by increasing begin position, and incrementally merge two adjacent regions $(b_k, e_k)$ and $(b_l, e_l)$ if $b_l < e_k$ and $e_l - b_k < \ell$.

### Identifying precise breakpoints

We identify precise breakpoints shared by all polymorphism carriers in order to exclude false positive insertion sites. Due to the biological mechanisms that lead to Alu insertions, the breakpoint is often not a single position. In the data, we observe target site duplications (TSDs) and deletions, as well as Alu insertions that we can only characterize at one end, cf. Fig. S1. A TSD is a sequence of 4–25 bp repeated just before and after the Alu element (*Deininger, 2011*). Also common are short deletions that accompany an Alu insertion. The most difficult cases are those where not only the Alu sequence is inserted, but also some novel sequence. For such compound insertions, we are typically able to identify a breakpoint for one end only.

Given an Alu insertion site, we define *AL* as the left breakpoint if there is a *C* read whose left part is mapped to the reference and whose right part is soft-clipped at *AL* and can be aligned to an Alu sequence. Similarly, we define *AR* as the right breakpoint of the Alu insertion if there is a *C* read whose right part is mapped to the reference and whose left part is soft-clipped at *AR* and can be aligned to an Alu sequence. *AL* is not always equal to *AR* and often only one of them can be characterized, as illustrated in Fig. S1. In our implementation we allow *AL* and *AR* to differ by up to 50 bp.

Given a region $(b_i, e_i)$, we declare that the region contains an Alu insertion if at least one of *AL* and *AR* can be unambiguously determined as described in the following. Otherwise, this region is excluded from further analysis. We use the term *breakpoint* to describe a pair $(AL, AR)$, or a single position *AL* or *AR*, when only one of the positions can be identified.

Ideally, all polymorphism carriers having *C* reads in this region will point to one single breakpoint. However, this is often not the case as some split reads are merely sequencing and/or mapping errors, indicating false breakpoints. This problem is partially solved by using only *good C* reads. For example, a *C* read most likely indicates a true breakpoint if it has good base calling quality and can be aligned to the reference with good alignment score. Nevertheless, the remaining *C* reads often suggest multiple positions.

To determine the true breakpoint, we introduce a two-level voting system, cf. Fig. 3. In this voting system, *AL* and *AR* are voted for independently of each other. Each region $(b_i, e_i)$ is processed as follows: (1) *C* reads from all individuals in this region are extracted and remapped to the reference using a split-mapping algorithm (*Emde et al., 2012*), which maps prefixes of the reads to the reference and the corresponding suffixes to an Alu sequence or vice versa. A set of 51 Alu sequences is provided with our code and we align to all provided sequences. The split position of each read in the alignment indicates a breakpoint on the reference genome. (2) The *C* reads of each individual are considered separately by the first level of the voting system. If the split alignment is successful, a *C* read adds its vote either for the respective left or right breakpoint position. For each individual the most common breakpoint position is determined and used in the next step. (3) The determined breakpoints from all individuals are collected as input to the second level of the voting system and the location with the highest number of votes is chosen.

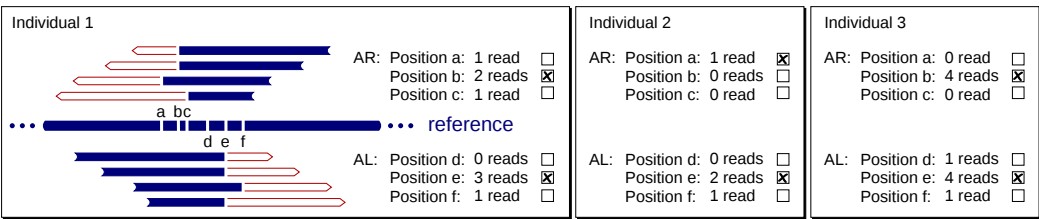

**Figure 3 Example instance of our two-level voting system that determines the exact breakpoints of an Alu insertion.** At the first level, split-reads vote for a left and right breakpoint position within each individual. At the second level, individuals vote for the positions that received the largest numbers of votes at the first level to choose the final breakpoint positions *AL* and *AR*. In this example, position *b* is elected as *AR* and position *e* is elected as *AL*.

### *Determining genotype*

If a breakpoint can be identified in a region $(b_i, e_i)$, we insert Alu sequences at the breakpoint in silico in order to adopt the Alu deletion algorithm for calling genotypes as follows. We consider the *D* and *C* read previously aligned to the set of Alu sequences as *I* and *S* reads, respectively. Further, we add $\delta$ (default $\delta = 300$) basepairs to the insert length distribution of reads spanning the insertion breakpoint, i.e., *A* reads. Using the three classes of reads, we can then apply the Alu deletion genotyping algorithm.

## Simulation of test data

In order to assess the accuracy of our approach, we simulated two sets of sequencing data, *SimDel* and *SimIns*, based on human chromosome 21 (build 37). In *SimDel*, we modeled recurring Alu deletions by selecting 100 known Alu elements on the reference chromosome 21, assigning a frequency to each of them, and deleting the Alu elements at these frequencies from 200 copies of chromosome 21, which resulted in 200 haplotypes. In *SimIns*, we modeled recurring Alu insertions by first deleting 100 known Alu elements from the reference chromosome 21, assigning a frequency to each of them, and inserting the Alu elements at these frequencies back into 200 copies of the modified reference, which again resulted in 200 haplotypes. In *SimIns*, we used the modified reference that has all 100 Alu elements deleted in all further steps of the analysis. In both sets, we chose the frequencies of the Alu element deletions or insertions to be uniformly distributed between 0 and 1. The selected Alu elements were chosen randomly from all known Alu elements on the reference chromosome 21, with the constraint that no other Alu is found within 600 bp of the inserted Alu.

Of every simulated Alu polymorphism, each individual can have 0, 1 and 2 copies, corresponding to the genotypes $G_0$, $G_1$ and $G_2$, respectively. $G_1$ represents a heterozygote and $G_2$ a homozygote Alu carrier. In our deletion data set *SimDel*, there are an average of 36.53 heterozygotes and 34.24 homozygotes per individual, and in the insertion data set *SimIns*, there are 37.53 heterozygotes and 33.66 homozygotes. See Table 1 for more details on the simulated Alu counts.

From the *Mason* read simulation package, version 2.0 (*Holtgrewe, 2010*), we used the *MasonVariator* to add SNPs and small indels to each haplotype and the *MasonSimulator* to

**Table 1  Simulated Alu counts of 100 individuals.** The *sum* column is the total counts of simulated Alu, the *min* and *max* column are the minimum and maximum number of Alu elements seen in one simulated individual.

| Dataset | $G_1$ | | | $G_2$ | | |
|---|---|---|---|---|---|---|
| | sum | min | max | sum | min | max |
| *SimDel* | 3,653 | 25 | 44 | 3,424 | 25 | 43 |
| *SimIns* | 3,753 | 26 | 47 | 3,366 | 26 | 44 |

generate paired-end Illumina reads for each haplotype (see Fig. S2 for details). We merged haplotypes in pairs to obtain 100 diploid data sets for *SimDel* and 100 diploid data sets for *SimIns*. For each *SimDel* and *SimIns* we simulated two read sets, one at an average coverage of ~10× and another at ~25×.

## Evaluation metrics

The advantage of a simulated data set is that we can measure accuracy by comparing the predicted genotype calls to the truth, including the accuracy of distinguishing heterozygous and homozygous calls. We count predictions per group $C_{tp}$, where $t \in \{0, 1, 2\}$ indicates the true genotype and $p \in \{0, 1, 2\}$ indicates the predicted genotype. Thus, $C_{tp}$ specifies the number of $G_p$ predictions where the true underlying genotype is $G_t$. For example, $C_{01}$ and $C_{02}$ count the number of false positives. We define the number of true positive calls (TPN) to tolerate genotyping errors, i.e., TPN $= C_{11} + C_{22} + C_{12} + C_{21}$, and calculate the sensitivity and false discovery rate (FDR) as

$$\text{Sensitivity} = \frac{\text{TPN}}{\text{TPN} + C_{10} + C_{20}} \quad \text{and} \quad \text{FDR} = \frac{C_{01} + C_{02}}{\text{TPN} + C_{01} + C_{02}}.$$

## RESULTS

In this section, we present our results of running PopAlu on both simulated data and on a human trio from the 1000 Genomes Project. We compare the results of PopAlu to RetroSeq (*Keane, Wong & Adams, 2013*) and Mobster (*Thung et al., 2014*), tools that are specialized for the discovery of transposable element insertions. As Mobster reports only the location of Alu insertions but no genotypes and cannot process the output of BWA mem to date, we focussed on the comparison to RetroSeq in most analyses.

## Simulated data

We ran PopAlu on *SimDel* independently for each individual, and on *SimIns* jointly for multiple individuals. Since RetroSeq does not report deletions, we ran it only on *SimIns*. Table 2 summarizes the predicted Alu calls for both data sets at both 10× and 25× read coverage.

As expected, the sensitivity of PopAlu increases on *SimDel* from 85.8% to 98.1% for the higher coverage data as more reads provide more information on the Alu polymorphisms. We observe a similar effect for on *SimIns*, although it is less pronounced than for *SimDel*.

**Table 2** **Summary of predicted Alu counts.** $C_{tp}$ represents the number of polymorphic Alu predicted as of genotype $p$ while the true underlying genotype is $t$. The counts of $C_{tp}$ are further grouped into 4 types, named as TP (True Positive), FN (False Negative), FP (False Positive) and GE (Genotype-calling Error). The definitions of Sensitivity and False Discovery Rate (FDR) are given in the main text.

| Coverage | Dataset | Tool | TP | | FN | | FP | | GE | | Sensitivity | FDR |
|---|---|---|---|---|---|---|---|---|---|---|---|---|
| | | | $C_{11}$ | $C_{22}$ | $C_{10}$ | $C_{20}$ | $C_{01}$ | $C_{02}$ | $C_{12}$ | $C_{21}$ | | |
| ~10× | SimDel | PopAlu | 2,668 | 3,350 | 985 | 19 | 0 | 0 | 0 | 55 | 85.8% | 0% |
| | SimIns | PopAlu | 3,152 | 3,017 | 601 | 341 | 0 | 0 | 0 | 8 | 86.8% | 0% |
| | SimIns | RetroSeq | 530 | 2,505 | 1,347 | 490 | 999 | 1,119 | 1,876 | 371 | 74.2% | 28.6% |
| ~25× | SimDel | PopAlu | 3,521 | 3,342 | 132 | 0 | 12 | 0 | 0 | 82 | 98.1% | 0.2% |
| | SimIns | PopAlu | 3,269 | 3,041 | 484 | 322 | 0 | 0 | 0 | 3 | 88.7% | 0% |
| | SimIns | RetroSeq | 2302 | 261 | 1,191 | 520 | 1,913 | 79 | 260 | 2,585 | 76.0% | 26.9% |

**Table 3** **Predicted and validated Alu Insertion calls for the CEU trio.** The *PCR (total)* column provides the total number of PCR validated Alu insertion calls by *Stewart et al. (2011)* for each sample and the *PCR* columns the number of validated calls that are also predicted by the program. The *Distance* columns show the average distance in basepairs between the predicted breakpoint and the breakpoint reported by PCR. For PopAlu, we calculated the distance from the mid-point of the reported interval. For Mobster, we calculated the distance based on the reported "Insert Point".

| Sample | PCR (total) | PopAlu | | | RetroSeq | | | Mobster | | |
|---|---|---|---|---|---|---|---|---|---|---|
| | | Total | PCR | Distance | Total | PCR | Distance | Total | PCR | Distance |
| NA12878 | 165 | 1,441 | 162 | 4.7 bp | 1,038 | 162 | 16.2 bp | 1,058 | 164 | 5.0 bp |
| NA12891 | 142 | 1,432 | 138 | 4.6 bp | 1,046 | 139 | 17.6 bp | 1,030 | 140 | 6.4 bp |
| NA12892 | 152 | 1,405 | 150 | 4.9 bp | 1,078 | 148 | 16.5 bp | 1,023 | 149 | 6.7 bp |

PopAlu performs consistently better than RetroSeq, as measured by sensitivity and FDR, with a much higher genotype calling accuracy.

## Real data from a 1000 genome project trio

We ran PopAlu with default parameters on public data of a CEU trio from the 1000 genome project (father NA12891, mother NA12892 and daughter NA12878). Within a follow-up study of the 1000 genomes pilot project (*Stewart et al., 2011*), the trio was sequenced at 9–16× coverage and 186 Alu insertion loci were randomly selected for PCR validation. We used high depth (>75×) Illumina HiSeq data generated at the Broad Institute,[1] the same data set used for assessing RetroSeq and Mobster. We compare our results to the set of PCR validated Alu polymorphisms (*Stewart et al., 2011*) and to results previously reported for RetroSeq (*Keane, Wong & Adams, 2013*) and Mobster (*Thung et al., 2014*).

As shown in Table 3, PopAlu reports about 35% and 38% more Alu insertions than RetroSeq and Mobster, respectively. On average PopAlu identifies 1,426 Alu insertions per sample, while RetroSeq and Mobster report on average 1,054 and 1,037. This is consistent with our results on simulated data, where PopAlu is more powerful than RetroSeq.

[1] Available at ftp://ftp.1000genomes. ebi.ac.uk/vol1/ftp/technical/working/ 20120117_ceu_trio_b37_decoy/.

**Table 4 Genotype calls of PCR validated Alu insertion calls for the CEU trio.** $C_{tp}$ represents the number of polymorphic Alus predicted as of genotype $p$ while the true underlying genotype is $t$. The true genotype was determined by PCR validation (*Stewart et al., 2011*).

| Sample | PopAlu | | | RetroSeq | | |
|---|---|---|---|---|---|---|
| | $C_{11}$ | $C_{22}$ | $C_{21}$ | $C_{11}$ | $C_{22}$ | $C_{21}$ |
| NA12878 | 124 | 38 | 0 | 124 | 1 | 37 |
| NA12891 | 95 | 41 | 2 | 95 | 0 | 44 |
| NA12892 | 107 | 41 | 2 | 106 | 0 | 42 |

Further, all three programs identify almost the same number of validated Alu insertions. Based on these numbers, the RetroSeq authors report an average sensitivity of 98% for each sample. However, the true number of novel Alu insertions is unknown, so the sensitivity may be inflated as it is presumably based on only a subset of all insertions. We further examined the accuracies in pinpointing the exact breakpoints. The mean distance from the true breakpoint is about 4.7 bp for PopAlu, about 6 bp for Mobster and about 17 bp for RetroSeq (see Table 3).

Next, we compared our genotype calls with the PCR validated calls (see Table 4). PopAlu has an average genotype calling accuracy of 99.1%, compared to an average of 72.6% for RetroSeq. The numbers show that RetroSeq has difficulties in distinguishing between homozygous and heterozygous carriers.

Finally, we counted non-Mendelian calls within the trio to estimate a lower bound on the FDR. Non-Mendelian calls are Alu insertions calls in the child that do not follow the expected inheritance patterns according to the calls in the parents. We include calls private to the child as false positives, since the rate of *de novo* Alu insertions is estimated to be about 1 per generation only (*Batzer & Deininger, 2002*). In the call set of PopAlu, we find in total 13 non-Mendelian calls, providing a lower bound to the FDR of 0.7%.

## Timing

We ran all computations on a desktop machine of a single 2.67 GHz Intel i5 processor. On average, the detection of Alu insertions with PopAlu took about 3 hs per sample from the CEU trio family.

## DISCUSSION

We have presented PopAlu, a method to detect and genotype polymorphic Alu insertions. The method can detect polymorphisms where the Alu element is present in the reference as well as where it is not. Our results indicate that the method has comparable or higher sensitivity to other tools and it can accurately distinguish between homozygous and heterozygous carriers. Further the evaluation suggests that the false discovery rate of PopAlu is low and that the precision in determining breakpoints is high.

Despite the positive results achieved, PopAlu can be improved and extended. First, the Alu sequence used in the Alu insertion genotyping algorithm is a representative Alu sequence; a better sequence may be determined by local assembly. Second, our algorithm

could additionally incorporate the length distribution of $I$ and $S$ reads. Third, PopAlu currently uses a greedy algorithm for finding insertion locations, it may benefit from the development of an optimal algorithm. In terms of extensions, PopAlu can be extended to find other types of retrotransposons, e.g., LINE elements. Further, we note that Alu elements are often inserted along with more sequence, which may possibly be detected by combining PopAlu with a local assembly approach, such as PopIns (*Kehr, Melsted & Halldórsson, 2015*). Finally, another future extension is the inclusion of somatic Alu insertion events.

## ACKNOWLEDGEMENTS

We thank Snædís Kristmundsdóttir for her comments on the manuscript and her help in maintaining the source code of PopAlu.

### Funding

A part of this work was done while Yu Qian was visiting deCODE genetics supported by NextGene (FP7-PEOPLE-2009-IAPP-251592). The funders had no role in study design, data collection and analysis, decision to publish, or preparation of the manuscript.

### Grant Disclosures

The following grant information was disclosed by the authors:
NextGene: FP7-PEOPLE-2009-IAPP-251592.

### Competing Interests

Birte Kehr and Bjarni V. Halldórsson are employees of deCODE genetics/Amgen.

### Author Contributions

- Yu Qian performed the experiments, analyzed the data, wrote the paper, prepared figures and/or tables.
- Birte Kehr analyzed the data, wrote the paper, prepared figures and/or tables, reviewed drafts of the paper.
- Bjarni V. Halldórsson conceived and designed the experiments, wrote the paper, reviewed drafts of the paper.

### Data Availability

Source code on GitHub:
https://github.com/aimeida/PopAlu.

### Supplemental Information

Supplemental information for this article can be found online at http://dx.doi.org/10.7717/peerj.1269#supplemental-information.

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
