# Peer review of "PopAlu: population-scale detection of Alu polymorphisms"

_PeerJ, doi:10.7717/peerj.1269_

## Round 0.1 · original submission · Minor Revisions

Both reviewers ask several questions about certain details of the paper, and I assume that they can be easily answered and the paper clarified accordingly. As for the comparison against Mobster, it would be interesting to do it or (if really not applicable) explain why not.

·

Basic reporting

Overall, the paper is very well written, and a pleasure to read.

I believe that the reader would appreciate two or three sentences on these questions:

What distinguishes Alu polymorphism discovery from deletion and insertion discovery in general?
What, consequently, would be the drawbacks of methods such as NovelSeq (Hajirasouliha et al.), MindTheGap (Rizk et al.) or PopIns for detecting Alu insertions?

Of course, Alu polymorphism discovery is special because of the existence of repeat databases, against which one can map the reads, but you do not really mention this in the Introduction.

A minor remark: in the first sentence of the abstract, what exactly do you want to communicate with 'combined'? Do you just mean 'together'?

Experimental design

In general, the experimental design is very neat. I appreciate in particular to transform Alu insertion discovery into an Alu deletion problem, by inserting potential insertions and to re-align, which looks novel to me (is it?)

A few minor things:

- What is the motivation/intuition behind the formula in line 117, page 4? [P(H_0|G_1)]
- page 5, Figure 2: it would be helpful if you could also illustrate la and ra reads, as described from line 136 on page 5.
- page 5, lines 143/144: What's the reasoning behind choosing these intervals (why exactly 3 sigma)?
- page 5: the procedures described on lines 150 till 163 (region forming) are irreproducible (hence do not adhere to PeerJ standards). What about a description of these heuristics in a supplementary file?
- page 5, lines 167/168: Figure 3 isn't really a helpful illustration of TSD's (and the corresponding issues)
- page 6, line 177: why should AL be equal to AR at all?

Validity of the findings

Overall, the results look sound and convincing. In general, I think that PopAlu makes a convincing tool!

One (somewhat bigger) question though: a comparison with Mobster is missing. We (as members of the Genome of the Netherlands consortium) believe that this is a state-of-the-art tool, possibly the best one currently available. If there no real reasons why a comparison does not make sense, please compare also with Mobster -- I argue that this is a necessity to really understand the value of your tool in terms of practical benefits.

Additional comments

See my comments above, they also should serve as general comments.

·

Basic reporting

No comments.

Experimental design

In general, the methods and experiments are described very clearly. However, there are three aspects where some additional details would aid understandability and reproducibility (without having to reverse engineer them from the provided software implementation, which is probably possible):

1) How does the heuristic partitioning mentioned on Page 5 (lines 155 and 163) work?

2) How is the consensus Alu sequence (mentioned on Page 6, lines 193 and 200) chosen? In my understanding, you use exactly ONE Alu sequence, right? Wouldn't it be helpful to use a collection, i.e. a complete "mobiome", as done by Mobster?

3) Page 6, Line 180: what exactly does "unambiguously determined" mean in this context?

Validity of the findings

- Making the simulated benchmark data available to the community would be helpful.

- Page 6: The description of SimIns is a bit ambiguous to me. You first say that the Alu elements where inserted back into the reduced reference (Line 208), presumably at the places they where deleted from, and then say that the Alu locations were chosen randomly (Line 212). Actually, wouldn't it be simpler to simulate just one data set and run it twice, once in "deletion/genotyping mode" using the full reference and once in "insertion/discovery mode" using a reduced reference? That would have the added benefit of showing whether performance correlates between the two experiments (e.g. some MEIs don't work in both settings because read mappability is poor in that region).

Additional comments

This is a nice paper that complements the yet rather sparse literature on mobile element discovery/genotyping. I've collected some minor comments/suggestions below:

- Discovery of deletions of known ALUs is the same thing as genotyping, right? I would point that out a bit more explicitly in the intro.

- In my opinion, referring to your H_0 and H_1 as "alleles" rather than "haplotypes" would fit better, because you are only talking about a single locus.

- Section read classification: Could be clarified a bit more. From the description of category "I" for instances, it is not immediately clear to me why r_8 belongs to I while r_6 does not. Is "is mapped to" (Line 98/99) equivalent to "the alignment overlaps with"?

- Page 4, Line 117: Motivating the choice of P(H_0|G_1) would be good.

- Page 6, Line 192: The grammar of this sentence seems odd.

- Page 6, paragraph on voting system: I think the manuscript could benefit from a figure explaining this voting system.

---

## Round 0.2 · accepted · Accept

The revisions resolve the questions or misunderstandings that the reviewers have had, and the paper is well written and readable.

·

Basic reporting

"The mentioned tools focus on insertions of novel sequences, sequences that are not present anywhere in the reference genome, unlike Alu and other repetitive sequences."

Yes, of course, I know all of those papers. My feeling was that it could be helpful to write one or two lines pointing out the differences between the challenges that Alu insertion discovery faces, and the challenges of novel (or also generic) insertion discovery.

Experimental design

"Three sigma (the 99.7 percentile) is commonly treated as near certainty in statistics (see for example Wikipedia -> ‘three-sigma rule’)."

Yes, of course. I knew the 3-sigma-rule even before Wikipedia existed. I also knew that you should be careful in applying it when applying it multiple times. I had formulated my question too sloppily.

I was wondering whether the 3-sigma threshold is a good choice also in view of applying this multiple times, hence you may have a (virtual) multiple testing issue here. In other words, I was asking whether 3 sigma was also a good choice in relation to the number of tests you perform. But in the meantime, I think that this does not play much of a role here, if you choose your support(p) threshold carefully -- and your method makes good results with the one you choose.
* * *
"In cases where there occurs no target site duplication, no deletion and no novel sequence insertion along with an Alu insertion, the insertion position for the left end of the Alu equals the insertion position for the right end, i. e. AL equals AR. In the Alu deletion problem AL is never equal to AR."

Thanks, I had had a misunderstanding about the definition of AL, AR.

Validity of the findings

One comment about Mobster: we are using it for genotyping (which works based on logistic regression), but this may not be publicly available yet.

·

Basic reporting

No Comments.

Experimental design

No Comments.

Validity of the findings

No Comments.

Additional comments

In the revised version, the authors have addressed all my concerns.